# Electromagnetic Waves in Cosmological Spacetime

Denitsa Staicova  and Michail Stoilov *

Institute for Nuclear Research and Nuclear Energy, Bulgarian Academy of Sciences, 1784 Sofia, Bulgaria;
dstaicova@inrne.bas.bg
* Correspondence: mstoilov@inrne.bas.bg

**Abstract:** We consider the propagation of electromagnetic waves in the Friedmann–Lemaître–Robertson–Walker metric. The exact solutions for plane and spherical wave are written down. The corresponding redshift, amplitude change, and dispersion are discussed. We also speculate about the connection of the electromagnetic wave equation to the Proca equation and its significance for the early Universe.

**Keywords:** Friedmann metric; electromagnetic waves; redshift; amplitude change; dispersion; Klein–Gordon equation; tachyon inflation

## 1. Introduction

Currently, most of the known information about the Universe outside the Earth is obtained observing electromagnetic waves. This is especially true for the deep cosmos, even though gravitational astronomy [1] and neutrino telescopes [2,3] are on the path of opening the doors of the multimessenger observations [4]. So far, however, the experiments that have shaped our vision about the Universe and its evolution, such as Planck [5], Wilkinson Microwave Anisotropy Probe [6], WiggleZ Dark Energy Survey [7], Background Imaging of Cosmic Extragalactic Polarization [8], All Sky Automated Survey for Super Novae [9], Sloan Digital Sky Survey [10], Hubble Space Telescope [11,12], James Webb Space Telescope [13], etc., all detect electromagnetic signals emitted somewhere faraway that reach us after a long journey through the spacetime. From them we know that the Universe is currently in the epoch of accelerated expansion [14] and that the flat Friedmann–Lemaître–Robertson–Walker (FLRW) metric [15] is, so far, the best candidate for large-scale metrics (and is considered as a standard model for cosmological observations).

In this work we consider the problem of how the electromagnetic waves propagate in the FLRW metric, which is quite important to observational cosmology. This is not a new problem and it has attracted constant attention during the past few decades [16–21]. A wide variety of methods has been used to solve it. Some of them (such as the usage of the Debye potential or the reformulation of the problem as propagation of the electromagnetic field in Minkowski space time in medium for which susceptibilities are determined by the metric tensor) are quite general and are suitable for different gravitational backgrounds. Other ones (such as using the conformal coordinates) are tailored specifically to the FLRW metric. Some of the authors work directly with the electric and magnetic fields; other ones prefer to work with the electromagnetic potential in different gauges (usual—the Coulomb one). Here our approach is to consider the problem using electromagnetic potential in the standard co-moving frame and in the generalized Lorenz gauge. We have three reasons which give grounds for our choice: the first two explain why we use the electromagnetic potential, while the third one answers the question of why we prefer the Lorenz gauge: First, the electromagnetic potential is the canonical coordinate in the electrodynamics, considered the Hamiltonian dynamical system, while the electric field is the canonical momentum and the magnetic field is derived from the potential. The eventual (canonical) quantization will rely deeply on the potential modes. Second, the Aharonov–Bohm effect demonstrates

the fundamental role of the interaction of the potential with charged particles. Third, the usage of a generally covariant gauge condition exhibits a very interesting peculiarity of the electrodynamics in curved spacetime: The strong equivalence principle states that the laws of physics are one and the same in all inertial frames. However, there is a result [22] that on a world line one can place any metric into Minkowski form (the inertial coordinate system) and to nullify its first derivatives but not the second and higher ones. As a result, any equation in which metric's second and higher derivatives appear is not the same in Minkowski spacetime and in local inertial coordinate system in a more general spacetime. The electrodynamics in curved spacetime (more specifically—in non-Ricci flat spacetime) is an example of such theory because of the presence of the Ricci tensor in its equation of motion.

We want to emphasize that, here, we treat the electromagnetic waves as test particles in a predefined metric (flat FLRW one in our case). In other words, we consider the electromagnetic field as a perturbation, i.e., we do not take into account the gravitation produced by the wave itself. In addition, we suppose the existence of a medium in the spacetime which, first, produces the FLRW metric, and second, is inert to the electromagnetic interaction.

The paper is organized as follows: We start with some basic facts about the free electromagnetic field in curved spacetime. We give the definitions of the electromagnetic field tensor, electromagnetic Lagrangian, equation of motion, and gauge fixing. In addition, we remind the reader of how the gauge fixing can be incorporated into the Lagrangian and write down the effective Lagrangian, corresponding to our setup.

Next, we consider two simple solutions of the free electromagnetic equation of motion—a plane wave and a spherical one. We show that in both cases under consideration, each physical (transverse) component of the potential decouples from other components, while the time and longitudinal components remain entangled. The equations for the transverse components are quite simple and can be solved exactly. Their solutions demonstrate relativistic redshift, amplitude decrease (fading), and dispersion.

We also speculate about the connection between electromagnetic wave equation in FLRW metric in the generalized Lorenz gauge and the Proca equation with mass determined by the Ricci tensor. It turns out that for a special case of equation of state, the electromagnetic wave describes a tachyon. Finally, we briefly discuss the possibility to use the metric-induced transition of photons and other $U(1)$ gauge field particles from tachyons to normal particles as a source of the inflation during the early Universe.

## 2. Theory

The electromagnetic potential $A^\nu$ defines the electromagnetic field tensor $F^{\mu\nu}$ which, due to its antisymmetry and the symmetry of Christoffel symbols ($\Gamma^\alpha_{\mu\nu} = \Gamma^\alpha_{\nu\mu}$), has the same form in any frame

$$F_{\mu\nu} = \nabla_\mu A_\nu - \nabla_\nu A_\mu = \partial_\mu A_\nu - \partial_\nu A_\mu. \tag{1}$$

Here, $\nabla^\mu$ is the covariant derivative in the spacetime. As a result, first, in any spacetime the electromagnetic field tensor is invariant under the usual gauge symmetry $A_\mu(x) \rightarrow A_\mu(x) + \partial_\mu f(x)$, where $f(x)$ is an arbitrary function, and second, the Lagrangian of the free electromagnetic field is exactly the same as in Minkowski spacetime:

$$\mathfrak{L} = -\frac{1}{4} F_{\mu\nu} F^{\mu\nu}. \tag{2}$$

The corresponding equation of motion is

$$\partial_\nu \left( \sqrt{-g} F^{\mu\nu} \right) = 0, \tag{3}$$

where $g$ is the determinant of the metric tensor $g_{\mu\nu}$. The above equation can be placed into explicitly covariant form, namely,

$$\nabla_\nu F^{\mu\nu} = 0. \tag{4}$$

Note that up to now we have not fixed the gauge freedom in the potential. In order to remove this arbitrariness we use the generalized Lorenz gauge

$$\nabla_\nu A^\nu = 0. \tag{5}$$

In this gauge, Equation (4) takes the form

$$\Box A^\mu - R^\mu{}_\nu A^\nu = 0. \tag{6}$$

Here, $\Box = \nabla^\nu \nabla_\nu$ is the d'Alembert operator defined for the covariant derivatives $\nabla^\mu$, and $R^\mu_\nu$ is the Ricci tensor

$$R_{\mu\nu} = \partial_\alpha \Gamma^\alpha_{\mu\nu} - \partial_\mu \Gamma^\alpha_{\alpha\nu} + \Gamma^\alpha_{\mu\nu} \Gamma^\beta_{\alpha\beta} - \Gamma^\beta_{\alpha\mu} \Gamma^\alpha_{\beta\nu}. \tag{7}$$

The following relations between the Riemann tensor $R^\nu_{\alpha\mu\nu}$ and Ricci tensor are used in the derivation of Equation (6):

$$(\nabla_\nu \nabla_\mu - \nabla_\mu \nabla_\nu) A^\nu = R^\nu_{\alpha\nu\mu} A^\alpha = R_{\alpha\mu} A^\alpha = R_{\mu\alpha} A^\alpha.$$

We want to stress that Equation (6) is not the wave equation ($\Box A^\mu = 0$) and that it cannot be cast into it with the help of change of coordinates. Therefore, if the spacetime is not Ricci flat then the equation satisfied by the electromagnetic waves will differ from the one in Minkowski spacetime even in the locally inertial coordinate frame.

The gauge condition can be incorporated into the Lagrangian. The easiest way to achieve this is with the help of the Lagrangian multiplier—a new auxiliary dynamical field $\lambda(x)$, so that the gauge fixed Lagrangian is

$$\mathfrak{L}_{\mathfrak{g}.\mathfrak{f}.} = -\frac{1}{4} F_{\mu\nu} F^{\mu\nu} - \lambda \nabla_\nu A^\nu. \tag{8}$$

There is a procedure, outlined, e.g., in Ref. [23], for how to integrate out the Lagrangian multiplier. Applying it, we obtain the following, fully covariant, gauge-fixed, effective Lagrangian

$$\mathfrak{L}_{\mathfrak{eff}} = -\frac{1}{4} F_{\mu\nu} F^{\mu\nu} - \frac{1}{2} (\nabla_\nu A^\nu)^2. \tag{9}$$

Note that in the last term of Equation (9) there is a part $\mathfrak{L}_2$ which is quadratic with respect to the potential $\mathfrak{L}_2 = A_\mu M^{\mu\nu} A_\nu$, $M^{\mu\nu} = \partial_\alpha(\sqrt{-g} g^{\mu\alpha}) \partial_\beta(\sqrt{-g} g^{\nu\beta})/g$, so Equation (9) resembles the Proca action (more precisely—$\mathcal{L}_4$ generalized Proca one [24]). We shall come back to this question later.

## 3. Results

Here, we describe some basic solutions of Equation (6) in the case of FLRW background. On the base of these solutions we give some estimates about the redshift, amplitude decrease, and dispersion which the electromagnetic wave undergoes during its propagation in spacetime.

### 3.1. Plane Electromagnetic Wave in the FLRW Metric

The description of a plane wave can be given most easily in the Cartesian form of the metric of a flat FLRW-universe. In this case the invariant length is

$$ds^2 = -dt^2 + a(t)^2 (dx^2 + dy^2 + dz^2), \tag{10}$$

where $a(t)$ is the scale "parameter" and we work in units where the speed of light $c = 1$.

For the sake of simplicity, we consider a wave propagating in the $z$ direction, i.e., we suppose that $A^\mu = A^\mu(t, z)$. We want to stress that in this way we can describe the most general plane wave solution because of the possibility to freely choose the spacial

coordinate system orientation. For the considered functional dependence of the potential, Equation (6) simplifies, and each of the potential transverse components $A^1$ and $A^2$ (i.e., $x-$ and $y-$components of the potential) decouple from all other components. Note that, here, the transverse potential determines transverse electric and magnetic fields. Therefore, we deal with a transverse electromagnetic wave which remains as such forever. Certainly, this is valid only in the coordinate system determined by Equation (10). For instance, in the observer frame, where $x_{ob} = a(t)x$ (and analogously for the other spacial coordinates), every component of the electromagnetic potential is a mix of physical and nonphysical degrees of freedom.

The equation, which both $A^1$ and $A^2$ satisfy is

$$-\ddot{A}^i + \frac{1}{a^2}A^{i\,\prime\prime} - 5\frac{\dot{a}}{a}\dot{A}^i - 2\left(2\frac{\dot{a}^2}{a^2} + \frac{\ddot{a}}{a}\right)A^i = 0, \ \ i = 1,2. \tag{11}$$

Here, we denote with dots the derivatives over time and with primes—derivatives with respect to $z$. The equations for the nonphysical components $A^0$ and $A^3$ are completely different and each of them entangles $A^0$ and $A^3$.

We look for a solution of Equation (11) with separated variables $A(t,z) = f(t) \times g(z)$. In this case, Equation (11) leads to the following two equations:

$$g'' + k^2 g = 0, \tag{12}$$
$$a^2\ddot{f} + 5a\dot{a}\dot{f} + (k^2 + 4\dot{a}^2 + 2a\ddot{a})f = 0, \tag{13}$$

where $k^2$ is some constant.

At this point, we want to make a small comment about Equation (12). The self-consistency of the assumption for separation of variables requires that $a(t)$ does not participate in it. In other words, the equation in question will have the same form for any $a(t)$, even for $a(t) = 1$. Thus, it will be the same as in flat Minkowski spacetime.

In view of the above comment, it is not a surprise that Equation (12) is well known. Its general real solution for $k^2 > 0$ (which we suppose) is

$$g(z) = \tilde{c}_1 \sin(kz) + \tilde{c}_2 \cos(kz) \tag{14}$$

where $\tilde{c}_i, \ \ i = 1,2$ are arbitrary real constants.

Now we move to Equation (13). In order to find its solution we make two consecutive ansatzes. First, let

$$f(t) = \frac{\mathfrak{f}(t)}{a(t)^2} \tag{15}$$

which leads to the following differential equation for $\mathfrak{f}(t)$:

$$a^2\ddot{\mathfrak{f}} + k^2\mathfrak{f} + a\dot{a}\dot{\mathfrak{f}} = 0. \tag{16}$$

The second ansatz is

$$\mathfrak{f}(t) = \mathbf{f}\left(\int^t \frac{d\tau}{a(\tau)}\right) \tag{17}$$

where the argument of the function $\mathbf{f}$ is the so-called conformal time and the differential equation for $\mathbf{f}$ is

$$d^2_\tau \mathbf{f}(\tau) + k^2\mathbf{f}(\tau) = 0. \tag{18}$$

Note that Equation (18) has the same form as Equation (12) and, accordingly, has the same type of solution. Therefore, the general solution of Equation (13) is

$$f(t) = \frac{1}{a(t)^2}\left(\bar{c}_1 \sin\left(k \int^t \frac{d\tau}{a(\tau)}\right) + \bar{c}_2 \cos\left(k \int^t \frac{d\tau}{a(\tau)}\right)\right). \tag{19}$$

### 3.2. Spherical Electromagnetic Wave in the FLRW Metric

We consider the spherical coordinate system as the appropriate one for description of spherical waves (provided it is positioned and oriented accordingly to the source and observer). In spherical coordinates the invariant length is

$$ds^2 = -dt^2 + a(t)^2 \left( dr^2 + r^2 d\theta^2 + r^2 \sin(\theta)^2 d\phi^2 \right) \tag{20}$$

and it determines the particular form of Equation (6) in this case.

We consider the electromagnetic wave propagation in $r$ direction, supposing that $A^\mu = A^\mu(t, r)$, and, in addition, supposing separation of variables. Therefore, we are looking for transverse potential

$$A^\perp = (0, 0, A^\theta(t, r), A^\phi(t, r)) = f(t)(0, 0, \mathbf{A}^\theta(r), \mathbf{A}^\phi(r)) \tag{21}$$

(the time dependence of both transverse components is indeed the same—see below).

Here, we see a flaw in our considerations because we postulate the existence of a transverse constant vector field on a sphere. However, such a field does not exist. Any transverse vector field on a sphere has to have at least one zero (or singularity). Therefore, one of our assumptions, or both of them, are incorrect. Nevertheless, we continue our analysis, supposing that it is valid at most only approximately in a small vicinity on the sphere for which we suppose resides at $\theta = \pi/2$, $\phi = 0$. This should be sufficient for a cosmological observer who, anyway, can gather information only locally.

It turns out that the evolution of each of the transverse components decouples from all other components as in the plane wave case. The corresponding equations of motion are

$$-\ddot{A}^\theta + \frac{1}{a^2}(A^\theta)'' - 5\frac{\dot{a}}{a}\dot{A}^\theta + 4\frac{1}{ra^2}(A^\theta)' - 2\left(2\frac{\dot{a}^2}{a^2} + \frac{\ddot{a}}{a}\right)A^\theta + \frac{1-b^2}{r^2a^2}A^\theta = 0, \tag{22}$$

$$-\ddot{A}^\phi + \frac{1}{a^2}(A^\phi)'' - 5\frac{\dot{a}}{a}\dot{A}^\phi + 4\frac{1}{ra^2}(A^\phi)' - 2\left(2\frac{\dot{a}^2}{a^2} + \frac{\ddot{a}}{a}\right)A^\phi = 0. \tag{23}$$

Now the primes denote derivatives with respect to $r$ coordinate and $b = \cot(\theta)$. The fact that Equations (22) and (23) are different is a surprise, taking into account that orientation of the coordinate system around the axis of observation is a matter of our choice.

Let us consider Equation (23) first. As it has been already mentioned we look for a solution with separated variables, so that $A^\phi(t, r) = f(t) \times g(r)$. It turns out that the differential equation for $f(t)$ thus defined coincides with Equation (13) considered in the plane wave case. For the equation satisfied by the function $g(r)$ we obtain

$$g'' + \frac{4}{r}g' + k^2 g = 0, \tag{24}$$

where, as in the case of plane wave, $k^2$ is some positive parameter. Its general solution is

$$\begin{aligned} g(r) &= -\frac{1}{kr}(\hat{c}_1 j_1(kr) + \hat{c}_2 y_1(kr)) \\ &= \frac{1}{(kr)^2}\left(\hat{c}_1\left(\cos(kr) - \frac{\sin(kr)}{kr}\right) + \hat{c}_2\left(\sin(kr) + \frac{\cos(kr)}{kr}\right)\right), \end{aligned} \tag{25}$$

where $j_n$ and $y_n$ are the spherical Bessel functions of first and second kind.

Next, we consider Equation (22) and we again look for a solution in the form $A^\theta(t, r) = f(t) \times g(r)$. Once again, the differential equation for $f(t)$ is exactly Equation (13). For the function $g(r)$ we obtain the following equation:

$$g'' + \frac{4}{r}g' + \left(k^2 + \frac{1-b^2}{r^2}\right)g = 0. \tag{26}$$

The general solution of Equation (26) is

$$g(r) = \frac{1}{r^{3/2}} \left( \tilde{c}_1 J_{\sqrt{5+4b^2}/2}(kr) + \tilde{c}_2 Y_{\sqrt{5+4b^2}/2}(kr) \right), \tag{27}$$

where $J_\alpha$ and $Y_\alpha$ are the Bessel functions of first and second kind. Note, however, that $\theta-$dependence reappears in Equation (27) through the quantity $b$. This contradicts our initial ansatz. Therefore, we reconsider our assumption and now we are looking for a solution in the form

$$A^\theta = \mathcal{A}^\theta(t,r) \sin(\theta). \tag{28}$$

Skipping the details, it is possible to show that the equation for $\mathcal{A}^\theta$ at $\theta = \pi/2$ is exactly Equation (23). This resolves the problem with rotational symmetry around observation axis. The result also rules out the possible polarization of the light induced by its propagation through the space which is suggested by Equations (25) and (27).

### 3.3. Cosmological Redshift, Fading, and Dispersion

In this section we consider some quantities that are of interest for the observers of the electromagnetic waves.

First, we define the redshift $z$. It can be derived in number of different ways in the cosmological context. Usually, one uses the geodesic of the photon to obtain the so-called cosmological redshift [25]. One can also solve the scalar wave equation for light with certain initial data [26,27]. One can also use the Einstein–Maxwell's equations in the spacetime defined with its metric to obtain the solutions and obtain it from there. This is the approach we will use in this paper, so the redshift is defined as

$$1 + z = \frac{\omega_e}{\omega_o} \tag{29}$$

where $\omega_e$ is the wave (angular) frequency at the moment $t_e$ and $\omega_o$ is the observed frequency at the moment $t_o$. Note that time dependence for spherical and plane waves is one and the same and is given by Equation (19). Therefore, in both cases the angular frequency at moment $t$ is determined by the wave period $\Delta t$, $(\omega_t = 2\pi/\Delta t)$ and for $\Delta t$ we have the following equation:

$$2\pi = k \int^{t+\Delta t} \frac{d\tau}{a(\tau)} - k \int^{t} \frac{d\tau}{a(\tau)} \approx \frac{k}{a(t)} \Delta t. \tag{30}$$

As a result, the frequency at the moment $t$ is $\omega_t = k/a(t)$, and so the redshift is

$$z_{\text{pl, sph}} = \frac{a(t_o)}{a(t_e)} - 1, \tag{31}$$

where subscripts stand for plane and spherical wave.

Next, we define the fading $\mathcal{F}$ as the amplitude decrease due to the propagation of the wave:

$$\mathcal{F} = \frac{\mathcal{A}_o}{\mathcal{A}_e}. \tag{32}$$

Here, $\mathcal{A}_e$ is the amplitude of the wave at the moment $t_e$ and $\mathcal{A}_o$ is the observed amplitude at the moment $t_o$, $t_o > t_e$.

Another interesting characteristic of the waves is their dispersion $\mathcal{D}$. Similar to the redshift, there are lot of different ways to define the dispersion; probably the simplest is an analogue of the group velocity dispersion: $\mathcal{D} = d^2\omega/d^2k$. However, in all cases under consideration here, the frequency is linear with respect to wave number, but the spatial part

of the wave phase is not in the case of spherical wave (see below). Therefore, our definition for dispersion is

$$\mathcal{D} = \frac{d^2 \text{arg}}{d^2 k},$$ (33)

where arg is the argument of the sine or cosine part of the wave function. The quantity is determined only at the moment $t_o$.

Applying the above definitions for both plane and spherical waves, we obtain the following results for the plane wave:

$$\mathcal{F}_{\text{pl}} = \left(\frac{a(t_e)}{a(t_o)}\right)^2,$$ (34)

$$\mathcal{D}_{\text{pl}} = 0.$$ (35)

The corresponding results for a spherical wave are different. Having in mind that $1/kr \ll 1$, we can use Equation (25) which defines the spacial-dependent component of transverse potential in the following form:

$$g(r) \approx \frac{1}{r^2}\left(\hat{c}_1 \cos(kr + \frac{1}{kr}) + \hat{c}_2 \sin(kr + \frac{1}{kr})\right).$$ (36)

We set the origin of the coordinate system at the geometric center of the spherical wave. We consider the propagation along the $r-$axis of a fixed phase of outgoing wave, so that

$$k \int^t \frac{d\tau}{a(\tau)} - kr - \frac{1}{kr} = \text{constant},$$ (37)

which can be used to determine $r_o$ from $r_e$, $t_e$, and $t_o$. Therefore, the fading and dispersion are

$$\mathcal{F}_{\text{sph}} = \left(\frac{r_e a(t_e)}{r_o a(t_o)}\right)^2$$

$$\approx \frac{a(t_e)^2}{a(t_o)^2 \left(1 + \frac{1}{r_e}\int_{t_e}^{t_o}\frac{1}{a} + \frac{1}{k^2 r_e^2}\right)^2},$$ (38)

$$\mathcal{D}_{\text{sph}} = \frac{2}{r_o k^3}.$$ (39)

### 3.4. The Photon Mass

The Ricci tensor in the FLRW metric is diagonal. Its space components $R_i^i$ (no summation over $i$), $i = 1, 2, 3$ (i.e., $x-$, $y-$, and $z-$ components in Cartesian coordinates and $r-$, $\theta-$, and $\phi-$ components in spherical coordinates) have one and the same value:

$$R_0^0 = \frac{3\ddot{a}}{a},$$ (40)

$$R_i^i = \frac{2\dot{a}^2 + a\ddot{a}}{a^2} = (2-q)H^2, \quad \forall i = 1, 2, 3,$$ (41)

where $q = -a\ddot{a}/\dot{a}^2$ is called, due to historical reasons, the *deceleration parameter* and $H = \dot{a}/a$ is the Hubble function. Note that $q$ is dimensionless, so the factor $2 - q$ is well defined. Note also that only the transverse components of the potential are physical. These, in the general case, are certain linear combinations of the spatial components, so we are interested in their mass only.

Except for the very first moments of the Universe's evolution, $R_i^i$ is a very slowly changing function of time and, therefore, the term $R_\mu^i A^\mu = R_i^i A^i$ (no summation over $i$) in Equation (6) behaves similar to a mass term and Equation (6) itself—similar to a variant of the Proca equation. The effective mass term for the physical degrees of freedom is zero

only for $a(t) = $ constant or for $a(t) = $ constant $\times t^{1/3}$. These are the only two cases for which Equation (6), rewritten in the inertial coordinate system, coincides with the wave equation of Minkowski electrodynamics. Note that the power law behavior of the scale factor ($a(t) \propto t^p$, $p = 2/(3(w + 1))$) is an exact solution of the Friedmann equations for the Universe full with pure fluid with constant equation of state $w = p/\rho$, where $p$ is the fluid pressure and $\rho$ is its density. In this more general case, the photon effective mass is

$$m_\gamma^2 \ (= R_i^i) = \frac{p(3p - 1)}{t^2} = \frac{2(1 - w)}{3t^2(w + 1)^2} \tag{42}$$

which is positive for $w < 1$. However, in the case $w > 1$, the mass squared is negative and the photon represents a *tachyon*.

It is interesting to see the photon mass for exponentially growing scale factor $a(t) \propto \exp(ht)$. In this case we obtain

$$m_\gamma^2 = 3h^2, \tag{43}$$

i.e., it is a positive constant (exactly).

It will be instructive to have some estimate for the magnitude of the effective mass that we are speaking about. Let us note that for the present-day values of the deceleration parameter and Hubble function $q \approx -0.55$, $H \approx 70$ km/s/Mpc the photon mass in an inertial coordinate system is $m_\gamma \approx 10^{-33}$ eV.

## 4. Discussion

The obtained closed expressions for the electromagnetic plane and spherical wave solutions allow us to estimate the observed redshift, fading, and dispersion. The obtained redshift (Equation (31)) coincides with the standard one known in the literature [15,28]. The amplitude decrease given in Equation (34) is referred to in [29] as "adiabatic". The extension to a nonflat Universe considered therein showed that $a(t)^{-1}$ decrease is possible. On the other hand, the amplitude decrease of a spherical wave (Equation (38)) demonstrates the dependence of the fading on the wave number. The fading is increasing when the wavelength is increasing. A result, connected to the above one, is given by Equation (39) and predicts nonzero dispersion of a spherical wave.

Finally, we want to make some comments about a possible connection between Equation (6) and the Proca equation. Here, we used the generalized Lorenz gauge to fix the gauge freedom. As a result, Equation (6) is not gauge-invariant. In the Proca case, the mass term breaks the gauge symmetry as well but it cannot be considered as a gauge-fixing term for the electromagnetic potential. Nevertheless, the equations of motion for both theories are quite similar. The main difference between them is that for the electrodynamics, in the generalized Lorenz gauge the effective mass of the photon is controlled by the metric tensor. In view of this we suppose that a solution of the Proca equation with suitably chosen mass can approximate a solution of Equation (6). We consider this fact, i.e., that the photons can be massive, as very interesting and the fact that they can be tachyons as even more interesting. Note that tachyonic models have a long history in cosmology. Some of them originate as special cases of k-essence theories with Dirac–Born–Infeld (DBI) action [30]. On the other hand, k-essence theories [31,32] are used to describe early inflation and dark energy through a minimally coupled scalar field with noncanonical kinetic term. In the tachyonic models [33,34], universe expansion (possibly accelerated) is produced while the tachyon rolls down towards its minimum. Tachyons have also been discussed in terms of the so-called tachyonic preheating [35], which may lead to explosive particle production. Ghost tachyons (i.e., models with negative sign of $\dot{\phi}^2$ in the Lagrangian) have been shown to cross the phantom line of the equation of state $w < -1$ [36]. How can our result be positioned in these studies? Note that our result is valid not only for electromagnetic field but for any $U(1)$ gauge field. For instance, it can be the fundamental $U(1)$ field of electroweak interaction which exists before spontaneous symmetry breaking. Suppose that in some early stage of the Universe the perfect fluid which determines the metric in

it has $w > 1$ equation of state (which can be achieved in some $k$-essence models [37–42]). In this case, the fundamental $U(1)$ gauge field becomes tachyonic and, in the spirit of the articles cited above, these tachyons can drive inflation. It will continue as long as $w > 1$, and during it the tachyons will roll to their energy minimum where they are infinitely fast, thus thermalizing a significantly larger than expected part of the Universe.

**Author Contributions:** Conceptualization, M.S.; methodology, M.S.; software, D.S. and M.S.; validation, D.S. and M.S.; formal analysis, D.S. and M.S.; investigation, D.S. and M.S.; writing—original draft preparation, M.S.; writing—review and editing, D.S. and M.S. All authors have read and agreed to the published version of the manuscript.

**Funding:** This research was funded by Bulgarian National Science Fund grant KP-06-N58/5.

**Data Availability Statement:** No new data were created or analyzed in this study. Data sharing is not applicable to this article.

**Conflicts of Interest:** The authors declare no conflict of interest.

## Abbreviations

The following abbreviation is used in this manuscript:

FLRW      Friedmann–Lemaître–Robertson–Walker

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
