# Peer review of "Electromagnetic Waves in Cosmological Spacetime"

_universe, doi:10.3390/universe9060292_

Round 1

Reviewer 1 Report

In this paper the authors explore the propagation of electromagnetic waves in cosmological spacetime geometry as described by the flat FLRW metric. It is a particular case of the theory of Maxwell fields in curved spacetime with relevance for cosmology. The paper is well written with clarity in the explanation of concepts and in the calculations. The topic of the paper is quite relevant for theoretical and observational cosmology and it also contributes for the study of the coupling between gravity and electrodynamics, where spacetime geometrical structures play a significant role.

I recommend this paper for publication with some minor revisions in the English writing and one or two suggestions: 

1) I understand the comparison that the authors make between eq (4) and the Klein-Gordon equation. Nevertheless, to be more rigorous it should be said that eq (4) is analogous to the Proca equation, the typical (second-order) differential equation for massive vector fields. If we are even more specific by comparing eq (4) with the Proca equation in Minkowski spacetime, then eq (4) resembles a generalized Proca equation because the D'Alembertian operator in curved spacetime applied to the 4-vector potential A introduces a term proportional to the first derivative of A, where the factor contracting with it can be put in the form of g*gamma (where g is the inverse metric and gamma the Levi-Civita connection). Now, in Proca theory the presence of the mass term explicitly breaks the U(1) gauge symmetry and a longitudinal mode appears besides the two transverse polarizations. It would be nice to see in this paper a discussion about the same type of breaking of the gauge invariance and the possibility of a longitudinal mode for (inertial) observers in the freely falling frame, since as correctly said in the paper the Ricci term in eq (4) cannot be set to zero in such frame of reference due to the second derivatives of the metric. Tidal forces are a signature of true gravitational fields experienced by freely falling observers and in the same way extra electromagnetic effects (in comparison to Maxwell theory in Minkowski spacetime) induced by the Ricci tensor can also be a signature of the gravitational interaction. Of course this is only true for non Ricci-flat spacetimes, physical situations where the Ricci tensor is non zero. That is the case of cosmological fluids in the early Universe and such effects could leave an imprint on the CMB. Also it would be interesting to see in this paper a quantitative discussion about the possibility of a longitudinal mode observed by the inertial observer in the freely falling frame. It is easy to verify that the condition for orthogonality (between wave 4-vector and the electromagnetic 4-vector field) cannot be obeyed by plane harmonic waves in all spacetime.  

2) In connection to 1), it would be nice to see the corresponding effective Lagrangian, from which an inertial observer could derive the above mentioned generalized Proca equation. That is, the effective generalization of Maxwell theory in Minkoski spacetime equivalent to the theory of Maxwell field in the curved cosmological spacetime. In that Lagrangian the geometrical terms (metric, its derivatives) can be explicitly developed into the corresponding FLRW expressions.  

As for the English, a revision is preferable and the following changes are recommended:

line     expression

21      "the problem of how ..."

22     "The same problem..."

30     "because of the presence of the Ricci..."

32      "Lorenz"  (not Lorentz gauge. Its a common mistake also in this paper along the text)

69    "remain as such forever..."

etc...

Author Response

We thank the referee for reading our manuscript and recommending it for publication.
According to the referee recommendation we have added some text into Introduction section. We hope that this improves the section  and makes it more informative. 
1. We thank the referee for the important remark about the Proca equation. We have added a text where we discuss the relation between our consideration and the (generalized) Proca model (lines 259–267).
A comment about  the intertwining of physical and nonphysical degrees of freedom in different frames has been added as well (lines 119--123).
2. We have written down the effective gauge fixed Lagrangian (eq. 9) and discuss its relation to the (generalized) Proca model (lines 100–102).
We have tried our best to improve our English.

Reviewer 2 Report

The authors ignore important previous results as in EPJC (2021) 81:908 where the problem of the Maxwell field on FLRW spacetimes is exhaustively studied at the classical and quantum levels. Therefore I cannot recommend publication.

Author Response

We thank the referee for the recommendation of  EPJC (2021) 81:908. It is indeed very important work on the topic and we included it as a citation. However, the solutions we discuss in our manuscript are not included in this work. 
According to the referee recommendation we have added some text into Introduction section. We hope that this improves the section  and makes it more informative, especially the paragraph where we explain why our approach is interesting and what is it relation to the ones, used in other articles on the subject (lines 22--50). 

Reviewer 3 Report

The authors study electromagnetic waves in Friedmann universes. The analysis involves the electromagnetic potential, rather than the electric and magnetic components of the Maxwell field. The authors provide analytic solutions is certain cases and also speculate about a possible connection between the EM wave equation and the Klein-Gordon equation. The paper appears technically sound to me and it is clearly written, so I recommend its publication.

Author Response

We thank the referee for reading our manuscript and recommending it for publication.

Round 2

Reviewer 1 Report

The paper is quite appropriate. Suggestion:

line 227

Instead of "They, in the general case"

it's better to write "These, in the general case"

Author Response

We want to thank the referee again for recommending our manuscript for publication. The referee’s suggestion on line 227 has been taken into account in the new version of our manuscript.